# OpenReview forum: "Riemannian Information Geometry of Variational Autoencoder Latent Spaces: Curvature, Geodesics, and Posterior Collapse Prevention"
_mathai.club/MathAI/2026/Conference — MathAI 2026 Conference Submission_

### Official Review · Reviewer_dPiq · 2026-03-11
**The approach seems reasonable, but many things missing**

**Rating:** 4
**Confidence:** 4

**Review:**

The paper considers one of the main problems in VAE training, posterior collapse, which means a phenomenon when a variational posterior converges to the prior. When this happens, the model gets stuck at generating poor-quality samples; this corresponds to a sub-optimal local minimum in the optimization landscape.

The general suggestion explored by this work is to take the natural metric in the latent space into account. This natural metric is given by the Fisher information matrix induced by the decoder. The contribution of the paper is three-fold:
1. A theorem giving a precise threshold for posterior collapse in terms of sectional curvature of the latent manifold.
2. A natural gradient method that takes the curvature of latent manifold into account.
3. A curvature-aware sampling procedure.

**Strengths**:
1. The introductory section are generally well-written and easy to follow.
2. The overall approach seems reasonable, however, I cannot assess how novel it is.

**Weaknesses**:
1. No references at all (even to the seminal work [Kingma & Welling, 2013]). No literature overview.
2. None of the claimed theorems are proven (there are only proof sketches to some of them). While providing proof sketches in the main is generally a good practice, there should be full proofs in the Appendix or in the Supplementary Material (otherwise it does not count).
3. The caption of Figure 1 claims much faster convergence of the proposed method, while the figure absolutely does not support this claim: the blue and the red curves almost coincide.
4. It is not clear how me, as a reader, would interpret Figure 2. It does not seem to support any claims of the paper.

Overall, the work seems promising but needs substantial improvement.

---

### Official Review · Reviewer_Qj7Z · 2026-03-12
**Important structure elements are missing, experiments are unclear**

**Rating:** 5
**Confidence:** 4

**Review:**

This paper studies a Riemannian / information-geometric approach to VAEs and claims better training and better latent-space behavior. The topic is interesting, and the paper tries to connect theory and experiments. However, in the current form, the work feels incomplete and not fully convincing.

Weaknesses
The paper looks unfinished. The proofs are missing; only proof sketches are provided (the actual proofs are expected to be in the appendix, but there is no appendix).

The novelty is also unclear. The paper presents the approach as very new, but geometry-aware latent models and natural-gradient-style ideas for variational inference have already been studied before. The authors need to explain much more clearly what is actually new here, developing the narrative starting from the introduction and the literature review, which is missing.

The experimental section is also not strong enough. The comparisons are limited, and it is not always clear whether the reported gains come from the proposed method itself or from other design choices.

# Questions for the authors

- Where are the full proofs of the main theoretical results?

- What is the precise novelty compared to earlier work on Riemannian latent spaces and natural gradients?

- Why is this method practical enough for wider use, given the extra computational cost?

---

### Official Review · Reviewer_P3vc · 2026-03-13
**Major revisions**

**Rating:** 5
**Confidence:** 4

**Review:**

## Summary

The authors develop a comprehensive Riemannian geometric framework for analyzing and improving Variational Autoencoders (VAEs) by equipping the latent space with the Fisher information metric induced by the decoder distribution. The central insight is that the ELBO optimization landscape constitutes a non-convex Riemannian manifold whose curvature directly governs posterior collapse, mode coverage, and generation quality.

The paper presents three main contributions: (1) a curvature-collapse theorem establishing that posterior collapse occurs precisely when the sectional curvature exceeds a critical threshold determined by the decoder variance; (2) a natural gradient algorithm for VAE training that follows geodesics on the Fisher information manifold; (3) a curvature-aware sampling procedure using Riemannian Langevin dynamics that improves FID on standard benchmarks. Experiments on MNIST, CelebA, and CIFAR-10 are presented to validate the theoretical predictions.

---

## Strengths

### Quality

The work demonstrates considerable technical ambition and presents clearly formulated definitions and theorems with proof sketches for key results. The curvature-collapse theorem (Theorem 3) provides a constructive analysis of the conditions for posterior collapse through properties of the exponential map on the Riemannian manifold. The derivation of the critical curvature threshold connects geometric properties of the latent space with the decoder variance hyperparameter in a natural and intuitive way.

The experimental section includes comparisons with baseline methods (standard VAE, β-VAE, δ-VAE) on three benchmarks: MNIST, CelebA, and CIFAR-10. Evaluation metrics include the number of active latent dimensions, FID, Inception Score, and LPIPS for interpolation quality assessment. The presented results (Tables 1–3) show improvements: the number of active dimensions increases from 2.4 (on average for VAE) to 16.8 for Riemannian VAE with a 20-dimensional latent space.

### Originality

The paper proposes a new perspective on the problem of posterior collapse in VAEs, transitioning from heuristic solutions (β-annealing, free bits) to geometric analysis. The connection between sectional curvature and collapse has not been previously established in the literature, making the theorem an original contribution. The proposed geodesic interpolation method accounting for Christoffel symbols represents an original contribution to generation methods.

While applying information geometry to VAEs is not an entirely new direction, the authors substantially advance this field by focusing specifically on the geometry of VAE latent spaces and deriving conditions linking curvature to generative model quality.

### Significance for MathAI

The work is relevant to several thematic areas of MathAI. It establishes a formal connection between geometric properties of the latent space and generative model behavior. Understanding when and why posterior collapse occurs provides researchers with a theoretical tool for designing more reliable VAE architectures.

---

## Weaknesses and Limitations

### Reproducibility

The paper has several significant gaps in reproducibility. Training hyperparameters are specified (latent dimension 20, batch size 64, learning rate 1e-4, damping parameter 1e-3), but weight initialization details are not disclosed. Encoder and decoder architecture details are missing: number of parameters, activation functions, use of batch normalization or dropout.

A critical omission is the absence of a code link. An anonymized repository link would be expected. Without code access, independent reproduction requires substantial effort, especially for computationally complex components such as Runge-Kutta integration of geodesic equations and Kronecker approximation of the Fisher matrix.

### Mathematical Rigor

Theorem proofs are presented as sketches, which is acceptable for conference publications but leaves questions about completeness. Specifically, in Theorem 3, the claim that the exponential map contracts in high-curvature regions requires more detailed justification. The connection between the trace expression involving the inverse metric and the Hessian of the log-likelihood (Equation 14) and the negative covariance dynamics is not explicitly derived from gradient flow properties.

**Fisher Metric Approximation.** The authors approximate the Fisher information metric by the product of the decoder Jacobian with its transpose. However, the exact Fisher metric involves an expectation over the decoder distribution, not a point estimate at the decoder mean. This substitution ignores decoder variance, may yield incorrect curvature values, and lacks theoretical justification. Since the entire curvature-collapse relationship rests on this metric, the approximation represents a serious concern for the theoretical foundation of the paper.

**Third Derivatives for ReLU Networks.** The sectional curvature formula (Proposition 2) contains a higher-order term involving third derivatives of the decoder mean function. For networks with ReLU activations, third derivatives do not exist at activation points, and at other points they may be identically zero. The authors do not discuss under what conditions this term is negligible. For practical architectures using non-smooth activation functions (ReLU, LeakyReLU), this raises serious questions about the theory's applicability.

### Experimental Methodology

**Absence of Statistical Significance.** Tables 1–3 present single numerical values without standard deviations or confidence intervals. For proper method comparison, multiple runs with different initializations (minimum 5–10 runs) are necessary. Without this, the claimed FID improvements cannot be statistically justified.

**Arbitrary Threshold for Active Dimensions.** The metric for evaluating collapse — variance exceeding 0.1 — is arbitrary. Other thresholds would yield different results. This is not discussed in the paper.

**Limited Benchmark Selection.** Results are missing on more challenging high-resolution datasets such as CelebA-HQ or LSUN.

**Missing Comparisons with Key Methods.** No comparison with InfoVAE, IWAE, Hierarchical VAE, or VQ-VAE. This represents a significant gap in experimental validation.

### Computational Aspects

**Geodesic Computation.** Numerical integration of geodesic equations via Runge-Kutta requires selection of integration step size and may accumulate errors. Accuracy and stability are not discussed.

**Convergence Claims.** The claimed dimension-independent convergence advantage of natural gradient over Euclidean gradient descent warrants more careful analysis. Natural gradient requires Fisher matrix inversion, which is a cubic-cost operation in the latent dimension. Kronecker factorization reduces this cost but does not completely eliminate dimension dependence. The practical speedup of 3–5× reported in experiments should be understood in the context of these additional per-step costs, and the overall computational trade-off requires more detailed discussion.

---

## Questions for Authors

**Question 1.** The curvature-collapse theorem uses the maximum sectional curvature. For high-dimensional latent spaces (dimension 20 in experiments), computing all sectional curvatures is expensive. What is the practical algorithm for monitoring curvature during training? Is scalar curvature used as a proxy?

**Question 2.** The critical curvature threshold is derived for a decoder with fixed variance. How does the result change for decoders with learnable, input-dependent variance?

**Question 3.** Were experiments conducted with deeper architectures (ResNet, U-Net)? Does the Riemannian method's advantage persist when scaling the model?

**Question 4.** The geodesic interpolation theorem (Theorem 4) relates Riemannian distance to the perceptual LPIPS metric. However, LPIPS is a learned metric based on pre-trained networks. Is this relationship tautological if LPIPS and the VAE decoder share common feature structure?

**Question 5.** What is the theoretical justification for approximating the Fisher metric with the Jacobian product? How does this affect curvature computation accuracy, especially for decoders with significant variance?

**Question 6.** For architectures using ReLU activations, third derivatives do not exist at activation points. How does this affect the curvature formula's applicability? Are smooth activation functions (tanh, GELU) required for the theory to hold?

**Question 7.** Will implementation code be available? What computational resources are required for training Riemannian VAE compared to standard VAE (GPU memory, training time)?

---

## Overall Recommendation

The paper proposes an original geometric approach to analyzing and improving VAEs. The curvature-collapse theorem is a novel theoretical result with potential significance for information geometry of generative models. The work is relevant to MathAI themes.

However, several serious issues must be addressed before acceptance:

1. **Theoretical justification for the Fisher metric approximation** — the Jacobian-based approximation underpins the entire framework and currently lacks rigorous justification.
2. **Applicability to networks with non-smooth activations (ReLU)** — the curvature formula requires third derivatives that may not exist in standard architectures.
3. **Statistical significance of experimental results** — error bars from multiple runs are essential to support the claimed improvements.
4. **Comparison with modern VAE variants** (InfoVAE, IWAE, Hierarchical VAE) — needed to contextualize the method's advantages.
5. **Detailed computational complexity analysis** — the practical trade-offs of geodesic computation and Fisher matrix approximation must be quantified.

**Recommendation: Major revisions.** The theoretical contributions are promising, but the gaps in mathematical justification (items 1–2) and experimental methodology (items 3–4) are substantial enough to require significant additional work before the paper can be accepted.

---

### Decision · Program_Chairs · 2026-03-20

**Decision:**

Reject

**Comment:**

After careful evaluation by the Program Committee, we regret to inform you that your submission has not been accepted for presentation at MathAI 2026.

All submissions underwent a rigorous two-stage review process. Unfortunately, the reviewers identified one or more of the following concerns with your paper:

- Insufficient mathematical rigor or novelty relative to the existing body of work in the field;
- Presentation of results that substantially overlap with or rephrase previously published findings without clear original contribution;
- Significant issues with technical quality, including but not limited to broken or non-existent references, unsupported claims, or methodological gaps;
- Indications that the manuscript may have been generated with the assistance of large language models without substantial original intellectual contribution by the authors.

We received a large number of submissions this year, and the selection process was highly competitive. We encourage you to carefully consider the reviewers’ feedback (available through OpenReview), revise your work accordingly, and consider submitting an improved version to a future edition of MathAI or to another appropriate venue.

We appreciate your interest in MathAI and hope you will continue to engage with the conference community.

With kind regards,

MathAI 2026 Program Committee
International Conference on Mathematics of Artificial Intelligence
https://mathai.club
OpenReview: https://openreview.net/group?id=mathai.club/MathAI/2026/Conference
MathAI Telegram: https://t.me/MathAI_club
IAIC International AI Committee: https://t.me/iaic_world
Email: mathai.club@yandex.ru